# The Alopecia Areata Phenotype Is Induced by the Water Avoidance Stress Test In *cchcr1*-Deficient Mice

**DOI:** 10.3390/biomedicines9070840

**Published:** 2021-07-19

**Authors:** Qiaofeng Zhao, Satoshi Koyama, Nagisa Yoshihara, Atsushi Takagi, Etsuko Komiyama, Akino Wada, Akira Oka, Shigaku Ikeda

**Affiliations:** 1Department of Dermatology and Allergology, Juntendo University Graduate School of Medicine, Tokyo 113-8421, Japan; zhao@juntendo.ac.jp (Q.Z.); s-koyama@juntendo.ac.jp (S.K.); nyoshiha@juntendo.ac.jp (N.Y.); t-attsu@juntendo.ac.jp (A.T.); etsuko-k@juntendo.ac.jp (E.K.); akwada@juntendo.ac.jp (A.W.); 2The Institute of Medical Sciences, Tokai University, Kanagawa 259-1193, Japan; oka246@is.icc.u-tokai.ac.jp; 3Atopy (Allergy) Research Center, Juntendo University Graduate School of Medicine, Tokyo 113-8421, Japan

**Keywords:** alopecia areata, CCHCR1, gene, knockout mice, animal model

## Abstract

We recently discovered a nonsynonymous variant in the coiled-coil alpha-helical rod protein 1 (*CCHCR1*) gene within the alopecia areata (AA) risk haplotype. We also reported that the engineered mice with this risk allele exhibited. To investigate more about the involvement of the *CCHCR1* gene in AA pathogenesis, we developed an AA model using C57BL/6N *cchcr1* gene knockout mice. In this study, mice (6–8 weeks) were divided into two groups: *cchcr1^−^*^/*−*^ mice and wild-type (WT) littermates. Both groups were subjected to a water avoidance stress (WAS) test. Eight weeks after the WAS test, 25% of *cchcr1^−^*^/*−*^ mice exhibited non-inflammatory foci of alopecia on the dorsal skin. On the other hand, none of wild-type littermates cause hair loss. The foci resembled human AA in terms of gross morphology, trichoscopic findings and histological findings. Additionally, gene expression microarray analysis of *cchcr1^−^*^/*−*^ mice revealed abnormalities of hair related genes compared to the control. Our results strongly suggest that *CCHCR1* is associated with AA pathogenesis and that *cchcr1^−^*^/*−*^ mice are a good model for investigating AA.

## 1. Introduction

Alopecia areata (AA) is a complex genetic and tissue-specific autoimmune disease characterized by nonscarring hair loss that may begin as patches that can coalesce and progress to cover the entire scalp and/or the whole body [1]. Most authors accept the hypothesis that AA is caused by a T-cell mediated autoimmune response targeting an unknown antigen in anagen-stage hair follicles [2,3].

Previous genome-wide association studies (GWAS) have implicated a number of immune and non-immune loci in the etiology of AA, but none have yet been demonstrated to be causative of the disease, and none have been functionally confirmed to be involved in AA pathogenesis [4,5]. Alleles of the human leukocyte antigen (HLA) genes within the major histocompatibility complex (MHC) locus on chromosome 6p21.3 have, so far, shown the strongest associations with AA across different ethnic groups [4,5]. The largest reported genome-wide meta-analysis of AA identified HLA-DRβ1 as a key etiologic driver [4]. However, no convincing susceptibility gene has yet been pinpointed in the MHC locus.

We recently discovered a nonsynonymous variant (rs142986308, p. Arg587Trp) in the MHC associated with AA susceptibility in *CCHCR1* (coiled-coil alpha-helical rod protein 1), which encodes a novel component of hair shafts. In addition, our results demonstrate that mice carrying this amino acid substitution display a patchy hair loss phenotype. We further identified keratin abnormalities in the hair shaft and observed differential expression of hair-related keratin genes, not only in alopecic mice, but also in hair follicles from AA patients with the risk variant. Therefore, our study identified a novel AA susceptibility variant that was validated by functional analysis [6].

The purpose of this study was to investigate the effect of *CCHCR1* gene deficiency on AA pathogenesis. First, we generated C57BL/6N mice with deletion of the *cchcr1* gene using Cre/loxP technology. As *cchcr1**^−^*^/*−*^ mice develop normally and exhibit no obvious behavioral or physical phenotypic defects, and because psychological stress is known to be a triggering factor in AA patients [7], the water avoidance stress (WAS) test, which is well-known to stimulate psychological stress in mice [8,9], was performed on *cchcr^−^*^/*−*^ and WT littermates.

## 2. Materials and Methods

### 2.1. Generation of cchcr1^−^^/^^−^ Mice

To generate the *cchcr1* gene “floxed” mouse, two loxP sequences were inserted into mouse *CCHCR1* locus by homologous recombination. To construct the targeting vector, a 2.8 kb mouse genomic fragment containing exon 2 and 3 of *CCHCR1* gene, a 0.9 kb fragment containing exon 4 and a 5.0 kb fragment containing exon 5–10 were amplified by PCR from RENKA ES cell genomic DNA [10]. The 0.9 kb genomic fragment was cloned between 2 loxP sites of the vector which contains Frt flanked PGK_neo cassette (phophoglycerate kinase 1 promoter driven neomycin resistant gene) as a positive selection marker and MC1_DTA cassette (polyoma enhancer/herpes simplex virus thymidine kinase promoter driven diphtheria toxin A gene) as a negative selection marker. Then, the 2.8 kb and 5.0 kb fragments were subcloned into the plasmid. Resulting targeting vector contains MC1_DTA cassette, 2.8 kb 5′ homologous arm, first loxP site, 0.9 kb floxed genomic region, frt flanked PGK_neo cassette, second loxP site and 5.0 kb 3′ homologous arm. This targeting vector was linearized and introduced into RENKA ES cells (C57BL/6N) by electroporation. After selection using Geneticin (Invitrogen, Carlsbad, CA, USA), the resistant clones were isolated, and their DNAs were screened for homologous recombinant by PCR using the following primer set: sc_5AF2: 5′-CAA CTG AGG CCC GTT ACA GAG-3′ and neo_108r: 5′-CCT CAG AAG AAC TCG TCA AGA AG-3′. PCR positive ES clones were expanded, and isolated DNAs were further analyzed by PCR amplification using following primer sets: sc_5AF2 and neo_108r for 5′ amplification, sc_3AR4: 5′-CGA GGC TCT TCC AGG CAA CAG-3′ and neo_100: 5′-ATC AGG ACA TAG CGT TGG CTA C-3′ for 3′ amplification, and lox_5_fw_6: 5′-TAG CCT CTG CTA TGT ACT G-3′ and lox_5_ar_1: 5′-AAG TCC CTA CTA AGC ACA CGT GG-3′ for amplification of first loxP region. Homologous recombination of these clones was also confirmed by genomic Southern hybridization probed with neomycin resistant gene.

Homologous recombinant ES cell clones were aggregated with ICR 8 cell embryos to generate chimeric mice. Germline transmitted heterozygous *cchcr1* floxed mice were obtained by crossing chimeric mice with a high contribution of the RENKA background with C57BL/6N mice. Targeted allele was identified by PCR with following primer sets: sc_5AF2 and neo_108r, and lox_5_fw_6 and lox_5_ar_1.

To obtain *cchcr1* knockout mice, *cchcr1* floxed mice were crossed with B6; CBA-Tg(CAG-Cre)47Imeg transgenic mice [11] to excise out exon 4 and neomycin resistant cassette. *Cchcr1* knockout mice without exon 4 were identified by PCR amplification using following primer set: 5A-F2: 5′-CTC ACC TAG AAT TCA GAC ATC C-3′ and 3A-R1: 5′-AGT TGC AAC TGG CTA TAG CTG C-3′. Homozygous *cchcr1* knockout mice were obtained by intercrossing of *cchcr1* heterozygous mice, and their genotype was determined with PCR amplification using following primer set: 5A-F2 and 3A-R1. *Cchcr1* knockout mice production was performed in TranGenic Inc. (Fukuoka, Japan) in accordance with the institutional guidelines.

The experimental protocol was approved by the Ethics Review Committee for Animal Experimentation of Juntendo University (Code: 260244, date of approval: 9 February 2015).

### 2.2. RT-PCR Analysis

cDNA was synthesized from 5 μg of DNase I–treated total RNA using Applied Biosystems 7500 Real-Time PCR Software (Life Technologies Co., Carlsbad, CA, USA). The specific primers for each gene were as follows: 5′-GCA GAA GAT GAG GTT GGA GAC T -3′ and 5′- CTT GTG CTT CGC CTC TTC CA -3′ for *CCHCR1* and 5′-TCC TTC TTG GGT ATG GAA TCC TG -3′ and 5′-GAG GTC TTT ACG GAT GTC AAC G-3′ for β-actin.

### 2.3. Western Blotting Analysis

The protein levels of experimental proteins in mouse skin tissues were determined by Western Blot. The total protein of skin tissues was extracted by RIPA buffer (Thermo Fisher Scientific, Waltham, MA, USA) with Protease inhibitor cocktail (Sigma, St. Louis, MO, USA), and BCA assay kit (Thermo Fisher Scientific, MA, USA) was performed to detect the protein concentration. Proteins were separated by 10% Tris-glycin SDS-PAGE (Bio-Rad, Hercules, CA, USA) under denaturing conditions and transferred to nitrocellulose membrane. After blocking with 5% skim milk in Tris-buffered saline for 1 h. The membrane was incubated with primary antibodies against *CCHCR1* (St John’s Laboratory, London, UK) overnight at 4 °C. The secondary antibody was added to the membrane and incubated for 1 h at room temperature. The Blot signal was developed by an enhanced chemiluminescence-detecting kit (Thermo Fisher Scientific, MA, USA). The Western Blot was proved for β-actin using a monoclonal antibody (1:2000; BioLegend, San Diego, CA, USA) as a loading control.

### 2.4. WAS Test

The stress procedure was performed according to previous methods, with slight modifications [9]. Mice were placed on a platform (10 × 10 × 8 cm) attached to the bottom of a plastic tank (45 cm length × 25 cm width × 25 cm height). The tank was filled with warm water (25 °C) to within 1 cm of the top of the block. The mice were placed on the platform to avoid the water stimulus for two hours. The test was repeated on each weekday for two weeks.

### 2.5. Morphological Analysis

Macroscopic findings and dermoscopic images were observed regularly. To prepare the sections for histological analysis, AA skin samples were fixed in formalin solution and then embedded in paraffin. The sections were stained with hematoxylin and eosin (H&E). For immunohistochemical analysis, the following primary antibodies were used: anti-rabbit CD4 (ab183685, Abcam, Cambridge, MA, USA) and biotin anti-mouse CD8a (BioLegend, San Diego, CA, USA). Tissue sections were counterstained with hematoxylin.

### 2.6. Electron Microscopy

For transmission electron microscopy, the skin samples were fixed in Karnovsky’s fixative and embedded in epoxy resin according to standard procedures. Ultrathin sections were prepared, stained with uranyl acetate and lead citrate, and observed with an HT7700 (HITACHI, Tokyo, Japan) transmission electron microscope. Magnification was 5000 folds. For scanning electron microscopy observation, the hair shafts were dehydrated in 100% ethanol. After coating with platinum, the samples were examined with an S-4800 field-emission scanning electron microscope (Hitachi, Tokyo, Japan). Magnification was 500 folds.

### 2.7. Total RNA Extraction

Frozen mouse skin tissue was homogenized with TRIzol reagent (Thermo Fisher Scientific, MA, USA) using zirconia beads. The tissue lysate was processed according to the manufacturer’s instructions. RNA samples were purified by an RNeasy MinElute kit (QIAGEN N N.V., Venlo, The Netherlands).

### 2.8. Microarray Analysis of Mouse Skin RNA

One hundred nanograms of total RNA was processed for use on the microarray using the GeneChip WT PLUS Reagent Kit (Thermo Fisher Scientific, MA, USA) according to the manufacturer’s instructions. The resultant single-stranded cDNA was fragmented, labeled with biotin, and then hybridized to the GeneChip Mouse Gene 2.0 ST Array. The arrays were washed, stained and scanned using the Affymetrix 450 Fluidics Station and GeneChip Scanner 3000 7G (Thermo Fisher Scientific, MA, USA) according to the manufacturer’s recommendations. The expression values were determined using Expression Console software, version 1.4 (Thermo Fisher Scientific, MA, USA) with the default robust multichip analysis parameters.

### 2.9. Statistical Analysis

Data were analyzed by Fisher’s exact test. Differences were considered significant at *p* < 0.05.

## 3. Results

First, we generated mice with deletion of *cchcr1* using Cre/loxP technology. The final recombined floxed allele is presented in Figure 1a *Cchcr1^−^*^/*−*^ mice were obtained by intercrossing heterozygous (*cchcr1^+/−^*) mice. After identifying Cre-mediated deletion and the inactivation of *cchcr1*, the expression of *cchcr1* was assessed by PCR (Figure 1b). We also analyzed the RNA and protein levels of *cchcr1* in the skin (Figure 1c,d). The *cchcr1^−^*^/*−*^ mice were born at the expected ratios and had normal weights. The mice developed normally, were fertile, and exhibited no obvious behavioral or physical phenotypic defects.

Next, a WAS test was performed on *cchcr1^−^*^/*−*^ mice (*n* = 48, male: 24, female: 24) and WT mice (*n* = 16, male: 8, female: 8) (both 6–8 weeks after birth). The WAS test was performed for two hours a day, five times per week, for two weeks.

Eight weeks after the WAS test, in contrast to WT mice (Figure 2c), approximately 25% of *cchcr1^−^*^/*−*^ mice spontaneously exhibited noninflammatory foci of alopecia in the dorsal area (Figure 2a and Table 1). Hair loss may also develop within a sharply demarcated, localized area or diffusely on the dorsal side. The skin of the hair loss area has a normal appearance. The hair-pull test showed an increase in dystrophic anagen hairs. Characteristic phenotypes of AA (short stubs of hair and exclamation mark hairs) were observed in the hair loss area by dermoscopy. We also identified black dots and tapering hairs as signs of AA (Figure 2a).

Eighteen weeks after the WAS test, the localized hair loss spontaneously recovered (Figure 2b). Regrowth began one to three months after hair loss and was followed by the recurrence of hair loss in the same or other areas. In addition, AA showed a variable prognosis in each mouse and sometimes recurred. In aging AA mice, intralesional hair regrowth was a highly characteristic feature of AA; the newly sprouted hairs were white and became thinner as they gradually approached the dorsum, and the majority of the hair follicles were catagen and telogen follicles.

The AA-like skin lesions in *cchcr1^−^*^/*−*^ mice induced by WAS were tested by HE staining, immunohistochemistry and scanning and transmission electron microscopy (SEM and TEM). The histopathologic appearance of AA mice included a variety of features. The follicles showed little inflammatory lymphocytic infiltration in the peribulbar region (Figure 3a). We also observed the infiltration of plasma cells and an increase in the number of hair follicles, as well as miniaturized anagen follicles. The bulbs of the hair follicles were infiltrated by CD4- and CD8-positive T cells. On the other hand, there was no infiltration of CD4- and CD8-positive T cells in the specimens from wild-type mice (Figure 3b).

SEM examination of skin from *cchcr1^−^*^/*−*^ mice revealed exclamation mark hairs and hair shaft abnormalities as well as the loss of the cuticle layer and exposure of the underlying cortex (Figure 3c). Dark cells and mast cells in a perifollicular location were observed by TEM examination (Figure 3d).

To investigate the biological functions underlying the observed hair loss in *cchcr1**^−^*^/*−*^ mice, we performed gene expression microarray analysis of dorsal skin biopsies of *cchcr1*
*^−^*^/*−*^ mice and compared them with that of wild-type mice as a control. This analysis identified 196 genes with up- or downregulation 4-fold or greater. Cluster analysis of the probes revealed that the upregulated gene cluster in *cchcr1 ^-/-^* mice included hair-related genes, such as the keratin and KRTAP genes (Figure 3e,f). Hair keratins and KRTAPs are the major structural components of the hair shaft and are specifically expressed in the medulla, cortex, and cuticle. Other upregulated genes included S100 calcium binding protein A3 (S100A3, substrate of PADI3), trichohyalin (Tchh, substrate of PADI3), peptidyl arginine deaminase type III (Padi 3, related to posttranslational modification) and homeobox C13 (Hoxc13, related to hair shaft differentiation). This result suggested that cchcr1 affects the hair shaft and its formation at the genetic level.

## 4. Discussion

We previously reported genome-edited mice with a nonsynonymous variant of the *CCHCR1* gene in the AA risk haplotype [6]. Among these mice, 2 (12.5%) of the 16 heterozygous mice and 15 (55.5%) of 27 homozygous mice showed patchy alopecia after birth without any stimulation (assessed for up to 10 months). Both genome-edited mice and AA patients with the risk allele displayed morphologically impaired hair growth and comparable differential expression of hair-related genes, including genes encoding hair keratin and keratin-associated proteins (KRTAPs) [6]. These results were consistent with the microarray analysis in this study; the upregulation of keratins and the KRTAP, Padi3, S100A3 and Hoxc13 genes was confirmed (Figure 3e,f), suggesting that cchcr1 deficiency caused very similar gene expression patterns in humans and mice with AA carrying risk variants in CCHCR1. In addition, in this study, SEM examination of hair from *cchcr1**^−^*^/*−*^ mice revealed exclamation mark hairs and hair shaft abnormalities. These findings strongly supported the hypothesis that the deficiency or dysfunction of *CCHCR1* is related, not only to the hair loss phenotype, but also to hair keratinization abnormalities. As far as we can explore, there is few papers that describe that the relationship between AA and structural fragility of hair. The pathology of AA is generally thought to be a collapse of the immune privileges of the hair follicle, but which is not known in detail. Hair fragility may involve as one factor that causes a collapse of the immune privilege.

Another finding that should be discussed is the presence of peribulbar lymphocytic infiltration in biopsy samples from human AA and mouse models of AA as this phenomenon is one of the markers that are useful for AA diagnosis. Peribulbar lymphocytic infiltration was obvious in the alopecic area of *cchcr1^−^*^/*−*^ mice (Figure 3a,b), but it was not observed in genome-edited mice. It is possible that the WAS test could induce peribulbar lymphocytic infiltration in *cchcr1^−^*^/*−*^ mice, but another possibility is that lymphocytic infiltration is not always observable in all alopecic skin lesions. In fact, Müller et al. reported that peribulbar lymphocytic infiltration and lymphocytic infiltration within the fibrous streamers could be seen in only 39% of horizontal sections of human AA skin samples [12].

*CCHCR1* is located 110 kb away from the HLA-C locus and is positioned between the *CDSN* and *SC1* genes [13], and it has also been thought to be involved in the pathogenesis of psoriasis in a transgenic mouse model [14]. Our study supported the involvement of *CCHCR1* in AA pathogenesis, but further experiments will be needed to prove the involvement of *CCHCR1* in psoriasis.

Additionally, past studies indicated that psychological stress might be a triggering factor in AA patients [7]. Therefore, to induce AA-like lesions in *cchcr1^−^*^/*−*^ mice, we utilized the “mild form” of the WAS test in this study. Briefly, the WAS test was performed for only two hours a day and was repeated five times per week for two weeks. The stress sessions were performed between 1000 and 1300 h to minimize the effects of the circadian rhythm [15]. Nonetheless, 25% of *cchcr1^−^*^/*−*^ mice spontaneously exhibited noninflammatory alopecia foci on the dorsal side. Grooming and barbering, which are well-known behaviors among stressed C57BL/6 mice, were not observed. The hair loss site was located on the back, not the face, which resulted in characteristic barbering behavior. The morbidity rate of *cchcr1^−^*^/*−*^ mice is higher than that (20%) of C3H/HeJ mice, which is a well-known model for AA [16]. This result strongly indicates that not only the WAS psychological stress test but also genetic factors (i.e., deficiency or nonsynonymous variants of *CCHCR1*) might be responsible for AA. This model also revealed unique clinical findings regarding the female predisposition toward AA (Table 1). A previous report indicated that the relative incidence of AA in one production colony of C3H/HeJ mice was 0.25% for female mice and 0.035% for male mice, but selective breeding raised the frequency to nearly 20% [16]. Aging C3H/HeJ mice are known to exhibit alopecia spontaneously. However, in this study, *cchcr1**^−^*^/*−*^ mice that underwent the WAS test developed hair loss at approximately 16 weeks of age. Therefore, the WAS test could trigger AA in nonelderly mice. The unequal induction of hair loss in female and male mice suggested that the AA that developed in mice shares many features with AA in humans.

In conclusion, we demonstrated for the first time that in *cchcr1* gene knockout mice in the C57BL/6N background, psychological stress can trigger AA-like lesions. Mice deficient in *cchcr1* are likely to be a good model in which to examine the pathophysiology of AA. The new AA mouse model enables us to not only further investigate the pathogenesis of AA but also explore new therapeutic strategies and test the therapeutic efficacy of a wide range of candidate treatments in preclinical studies.

Further research may provide great clinical and therapeutic significance.

## Figures and Tables

**Figure 1 biomedicines-09-00840-f001:**
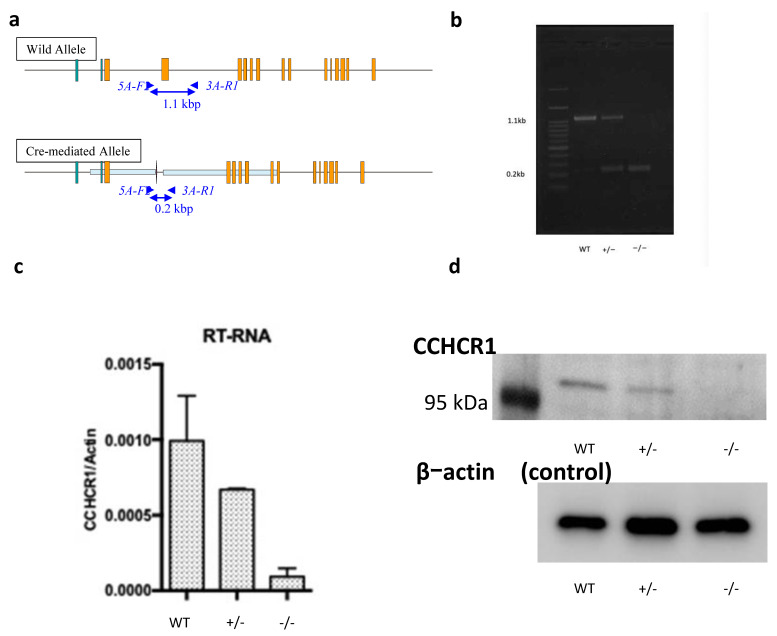
Generation of *cchcr1^−^*^/*−*^ mice. (**a**) Exon 4 was deleted through the loxP site in *cchcr1*. (**b**) Genotypes were identified by PCR of tail tips. The samples were loaded as follows: first lane, molecular weight marker; the WT band is at 1.1 kbp; the two *cchcr1^+/−^* bands are at 1.1 kbp and 0.2 kbp; and the *cchcr1^−^*^/*−*^ band is at 0.2 kbp. (**c**) The expression of *cchcr1* in WT mice, *cchcr1^+/−^* mice, and *cchcr1^−^*^/*−*^ mice was examined by RT-PCR. β-actin was used as an internal control. (**d**) Western blot showing the expression of *cchcr1* in skin tissue lysates from WT, *cchcr1^+/−^* and *cchcr1^−^*^/*−*^ mice.

**Figure 2 biomedicines-09-00840-f002:**
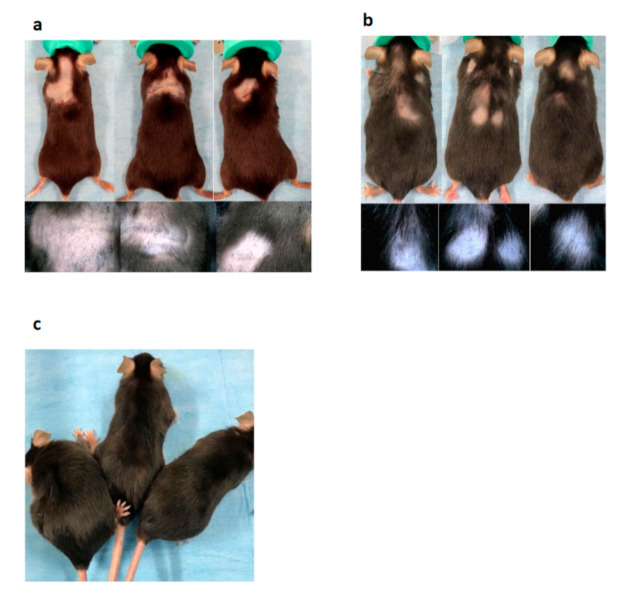
Macroscopic findings and dermoscopic findings. (**a**) Eight weeks after the WAS test, 25% of *cchcr1^−^*^/*−*^ mice showed localized hair loss. Short stubs of hair and exclamation mark hairs were seen under dermoscopy. (**b**) Eighteen weeks after the WAS test on *cchcr1^−^*^/*−*^ mice. Recovery from localized hair loss was evident, but the prognosis was variable, and AA-like lesions recurred in some mice. Thin hairs and white hairs were observed under dermoscopy. (**c**) WT mice after the WAS test.

**Figure 3 biomedicines-09-00840-f003:**
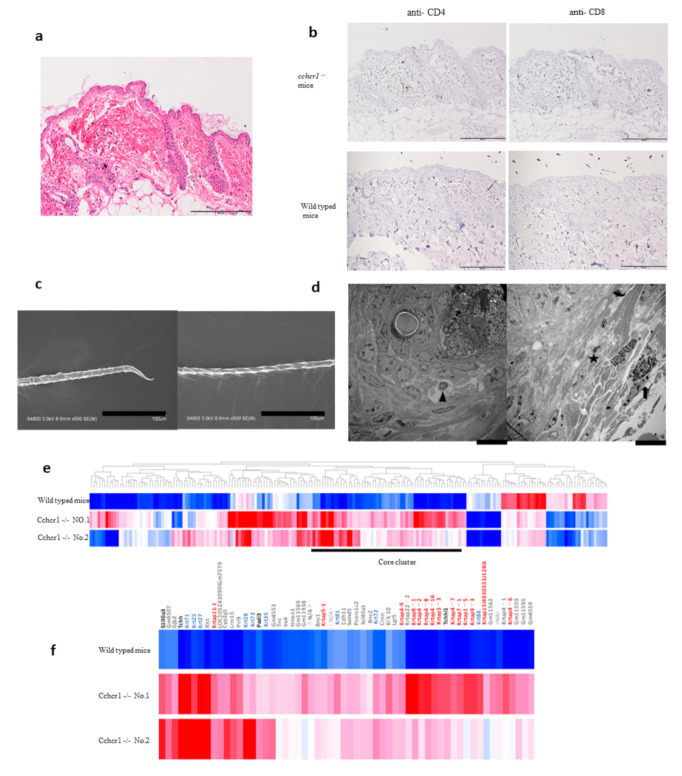
Morphological changes and genetic analysis. (**a**) Histological analysis of alopecic lesions in *cchcr1^−^*^/*−*^ mice using HE staining revealed follicles with little inflammatory lymphocytic infiltration in the peribulbar region (HE staining; scale bar = 200 µm). (**b**) IHC staining of the skin using CD4+ and CD8+ antibodies showed peribulbar lymphocytic inflammation in alopecic lesions in *cchcr1^−^*^/*−*^ mice (IHC staining; scale bar = 200 µm). (**c**) SEM examination of the hair shafts of *cchcr1^−^*^/*−*^ mice. Exclamation mark hairs (left panel) and abnormal hair shaft formation (right panel) were observed (scale bar = 100 µm). (**d**) TEM examination of the skin from alopecic lesions on *cchcr1**^−^*^/*−*^ mice revealed lymphocytes, dark cells, and mast cells in a perifollicular location (left panel). Apoptosis in the outer root sheath of catagen follicles, matrix keratinocytes and anagen hair bulbs was observed (right panel) (Scale bar = 10 µm). (**e**) Expression analysis of *cchcr1 ^-/-^* mice. Heat map of 196 probes showing ≥4-fold change in gene expression. (**f**) Heat map of core cluster genes. The color code depicts KRTAP family (red), keratin family (blue), other hair-related (black), and nonhair-related (gray) genes.

**Table 1 biomedicines-09-00840-t001:** The numbers of mice with AA-like skin lesions after the WAS test.

Observation Period after WAS		0 Weeks after WAS	5 Weeks after WAS	7 Weeks after WAS	9 Weeks after WAS	17 Weeks after WAS	Total
KO mice (*n* = 48)	Female (*n* = 24)	1	1	5	3	1	11
	Male (*n* = 24)	1	0	0	0	0	1
WT mice (*n* = 16)	Female (*n* = 8)	0	0	0	0	0	0
	Male (*n* = 8)	0	0	0	0	0	0

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
