# Peer review of "The Alopecia Areata Phenotype Is Induced by the Water Avoidance Stress Test In cchcr1-Deficient Mice"

_biomedicines, 2021, doi:10.3390/biomedicines9070840_

Round 1
Reviewer 1 Report
This is an interesting report. However in the manuscript there are no informations about clinical implications of their investigation. Accordingly the authors should add also a little paragraph about the possible clinical and therapeutic implications of their research.
Author Response
Response: Thank you for your comment. We have added the sentence in the conclusion.
Page 16 line 283:
Insert: Further research may provide great clinical and therapeutic significance.
Reviewer 2 Report
Zhao et al. report the results of a water stress test on hair loss in mice lacking cchcr1. In their previous work, they demonstrated hair loss in a mouse carrying a point mutation whereas here they describe a knockout model of the alopecia areata associated gene. They show that knockout mice, nearly all female, experience hair loss after the water stress test and conclude that the model phenocopies the disease in humans. They further offer that this model may be used for the study of this disease. My minor comments are below:
The resolution of the figures is low and must be improved. Some are quite difficult to read.
The number of female versus male mice should be included in table 1. As it stands, it is assumed in the table and the discussion that there is a sex effect in this knockout model; however, the reader is left in the dark in regard to the number of female versus male knockouts that developed hair loss after the stress test.
The methods are mostly well described but the magnification or resolution used for microscopy should be clearly stated in this section. It does not escape my attention that the scale is included in the figure legend.
Author Response
We appreciate the valuable comments from reviewer. We have addressed the reviewer’s questions and comments as follows:
- The resolution of the figures is low and must be improved. Some are quite difficult to read.
Response: Thank you for your comment. The parts that were difficult to see due to low resolution were not closely related to the text, so they were deleted and the figure was corrected.
- The number of females versus male mice should be included in table 1. As it stands, it is assumed in the table and the discussion that there is a sex effect in this knockout model; however, the reader is left in the dark in regard to the number of females versus male knockouts that developed hair loss after the stress test.
Response: In accordance with your suggestion, we have corrected the table 1.
- The methods are mostly well described but the magnification or resolution used for microscopy should be clearly stated in this section. It does not escape my attention that the scale is included in the figure legend.
Response: In accordance with your suggestion, we have added the sentences about magnification in the methods.
Page 9 line 148, 151:
Insert: Magnification was 5000 folds. / Magnification was 500 folds.
